# Influence of Polymorphism on the Electrochemical Behavior of Dilithium (2,3-Dilithium-oxy)-terephthalate vs. Li

Lou Bernard [1,2], Alia Jouhara [1], Eric Quarez [1], Yanis Levieux-Souid [3], Sophie Le Caër [3], Pierre Tran-Van [2], Stéven Renault [1] and Philippe Poizot [1,*]

1 Institut des Matériaux de Nantes Jean Rouxel, CNRS, IMN, Nantes Université, 44000 Nantes, France; lou.bernard@cnrs-imn.fr (L.B.); alia.jouhara@cnrs-imn.fr (A.J.); eric.quarez@cnrs-imn.fr (E.Q.); steven.renault@cnrs-imn.fr (S.R.)
2 Technocentre Renault, 1 Avenue du Golf, 78280 Guyancourt, France; pierre.tran-van@renault.com
3 NIMBE, UMR 3685 CEA, CNRS, Université Paris-Saclay, CEA Saclay, 91191 Gif-sur-Yvette, France; yanis.levieux-souid@cea.fr (Y.L.-S.); sophie.le-caer@cea.fr (S.L.C.)
* Correspondence: philippe.poizot@cnrs-imn.fr

**Abstract:** Organic electrode materials offer obvious opportunities to promote cost-effective and environmentally friendly rechargeable batteries. Over the last decade, tremendous progress has been made thanks to the use of molecular engineering focused on the tailoring of redox-active organic moieties. However, the electrochemical performance of organic host structures relies also on the crystal packing, like the inorganic counterparts, which calls for further efforts in terms of crystal chemistry to make a robust redox-active organic center electrochemically efficient in the solid state. Following our ongoing research aiming at elaborating lithiated organic cathode materials, we report herein on the impact of polymorphism on the electrochemical behavior of dilithium (2,3-dilithium-oxy-)terephthalate vs. Li. Having isolated dilithium (3-hydroxy-2-lithium-oxy)terephthalate through an incomplete acid-base neutralization reaction, its subsequent thermally induced decarboxylation mechanism led to the formation of a new polymorph of dilithium (2,3-dilithium-oxy-)terephthalate referred to as Li$_4$-$o$-DHT (β-phase). This new phase is able to operate at 3.1 V vs. Li$^+$/Li, which corresponds to a positive potential shift of +250 mV compared to the other polymorph formerly reported. Nevertheless, the overall electrochemical process characterized by a sluggish biphasic transition is impeded by a large polarization value limiting the recovered capacity upon cycling.

**Keywords:** lithiated organic materials; organic battery; polymorphism; dihydroxyterephthalate

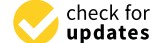

## 1. Introduction

Lithium-Ion Batteries (LIBs) are well-known to dominate the field of electrochemical storage technologies from portable-electronic market to electric vehicles. They obviously meet the requirements in terms of energy density, cyclability, and safety, but they also bring increasing challenges regarding scarcity of resources [1–3] and recyclability [4,5] due to the emerging large-scale application markets as highlighted by our group for several years [6]. Among the various alternative chemistries, the use of electrodes made of electroactive organic compounds could limit the demand on metal-based raw materials as well as mitigate the environmental burden of current LIBs [7,8]. Indeed, interesting assets like the abundancy of their constitutive atoms all over the world, their mild synthesis conditions or simpler recycling solutions can be put forward in their favor. In the past fifty years many promising organic electrode materials have been identified for possible applications in various cell configurations giving rise to an abundant and very interesting literature on the topic (see for example Refs. [7,9–16]). Note that pioneering works were reported in this field as early as 1997 by M. Armand and co-workers by demonstrating the possible Li insertion in oxocarbons to reach specific capacities higher than 500 mAh g$^{-1}$ [17], paving the way for further studies on lithiated discrete molecules.

For the last decade, our group has been focusing on lithiated positive organic electrodes especially by working on the robust and efficient dihydroxyterephthalate redox-active backbone (Figure 1). We first reported dilithium (2,5-dilithium-oxy)-terephthalate (Li$_4$-$p$-DHT or (Li$_2$)(Li$_2$)-$p$-DHT) as lamellar host electrode material [18], which displays an interesting stability upon cycling at an average potential of 2.55 V vs. Li$^+$/Li although limited to half of the theoretical capacity in our experimental conditions. Its full capacity (i.e., 2-electron reaction; 223 mAh g$^{-1}$ at 0.1 C) was later demonstrated by Chen group when prepared as graphene-supported nanosheets [19,20]. The study of its *ortho* regioisomer, namely, dilithium (2,3-dilithium-oxy)-terephthalate (Li$_4$-$o$-DHT or (Li$_2$)(Li$_2$)-$o$-DHT), enabled a potential gain of +300 mV reaching 2.85 V vs. Li$^+$/Li [21] thanks to the mitigation of stereoelectronic effects at the molecular level [22]. More recently, keeping the lamellar arrangement in the solid state, the cation substitution on the carboxylate functional groups was also found to be highly effective to tune the redox potential of the lihium phenolate redox-active moiety [23,24]. Thus, the reversible electrochemical lithium extraction occurs at 3.45 V vs. Li$^+$/Li in magnesium (2,5-dilithium-oxy)-terephthalate (Mg(Li$_2$)-$p$-DHT) due to the high ionic potential value of Mg$^{2+}$ ion that influences the Li–O bonds causing changes in the electronic distribution in the ligand (stabilization of the distribution of p-electron density) [23].

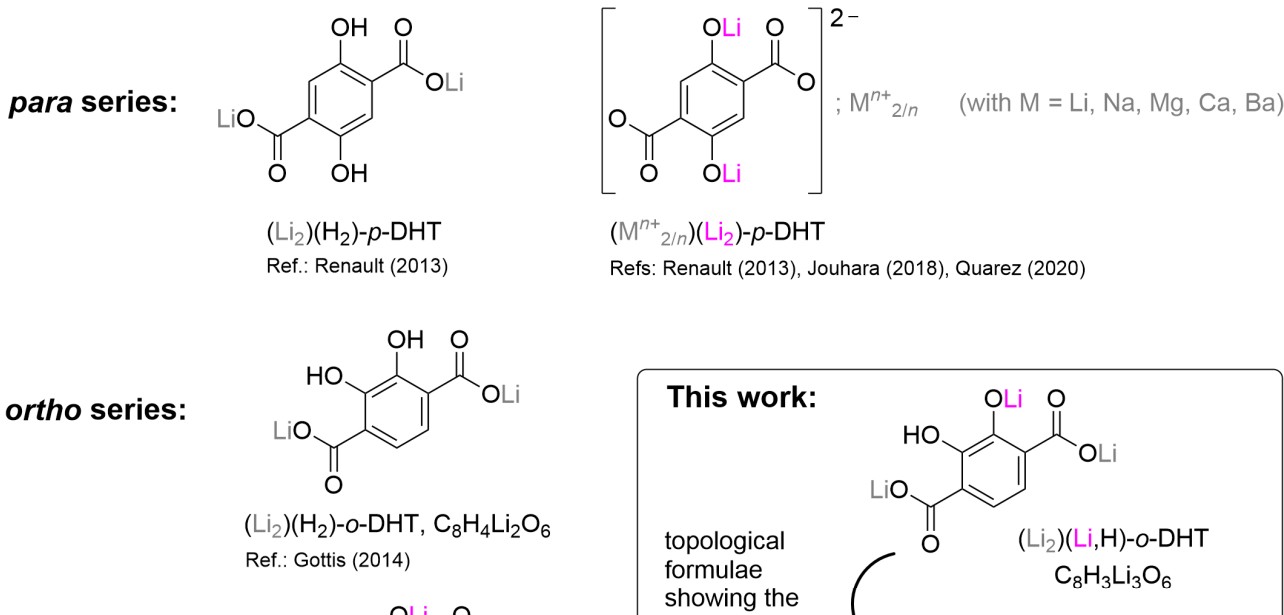

**Figure 1.** Topological chemical formulae of previously synthesized dihydroxyterephthalate (DHT) derivatives by our group together with the new compound reported in this article. Note that Li$^+$ cations potentially involved in a cathodic process are in pink.

Recently, complementary investigations performed on the *ortho*-regioisomer backbone led us to synthesize an intermediate compound, namely, dilithium (3-hydroxy-2-lithium-oxy)terephthalate or (Li$_2$)(Li,H)-$o$-DHT (Figure 1). Unexpectedly, after subsequent thermal treatment in neutral atmosphere, a polymorph of the previously described (Li$_2$)(Li$_2$)-$o$-DHT (or Li$_4$-$o$-DHT) compound [21] was recovered by intermolecular thermal rearrangement

(retro-Kolbe-Schmitt-type reaction). In this context, we report herein this complementary study focused on the *ortho*-regioisomer backbone by explaining the key chemical reactions leading to the synthesis of $(Li_2)(Li,H)$-*o*-DHT and $(Li_2)(Li_2)$-*o*-DHT (new polymorph), respectively. The electrochemical behavior measured in Li half-cell for the latter are also described and compared to data reported earlier [21].

## 2. Results

### 2.1. Preliminary Observations

In a former study [21], we reported, for the first time, the synthesis of the $(Li_2)(Li_2)$-*o*-DHT material performed in solventless conditions thanks to the occurrence of the retro-Kolbe–Schmitt reaction by controlled heating under inert atmosphere of the dilithium dicarboxylate $(Li_2)(H_2)$-*o*-DHT precursor. The latter was obtained by acid-base reaction when adding lithium carbonate in an aqueous suspension of 2,3-dihydroxyterephthalic acid ($H_4$-*o*-DHT). To complete this pioneer study, we recently tried to synthesize $(Li_2)(Li_2)$-*o*-DHT by direct four-proton neutralization in methanol (MeOH)/lithium methoxide (MeOLi) medium as previously reported by Chen group for the preparation of the *para*-regioisomer, namely, $(Li_2)(Li_2)$-*p*-DHT [20]. Note that such a chemical approach was also successfully applied by our group to synthesize the tetralithium salt of tetrahydroxybenzoquinone (THQ, 2,3,5,6-tetrahydroxy-1,4-benzoquinone) in 2009 [25]. In practice, 4.4 equivalents of MeOLi in methanol were added dropwise to a suspension of $H_4$-*o*-DHT and subsequently stirred overnight; the solid phase was then recovered by filtration. However, several analyses performed on the as-obtained powder led to the conclusion that an incomplete lithiation reaction systematically occurred by producing the $(Li_2)(Li,H)$-*o*-DHT) compound (Scheme 1), which is an intermediate between the $(Li_2)(H_2)$-*o*-DHT and $(Li_2)(Li_2)$-*o*-DHT.

**Scheme 1.** Summary of the neutralization reaction of 2,3-dihydroxyterephthalic acid ($H_4$-*o*-DHT) in presence of 4.4 equivalents of MeOLi in MeOH showing that a partial neutralization reaction occurs.

Although methoxide ion is a strong base, this partial neutralization of the catechol moiety can be explained by the existence of an intramolecular hydrogen bond once the first proton is removed (Figure 1), which does not exist in the *para*-regioisomer. Therefore, the remaining proton is better stabilized. This induces an increase of the $pK_{a2}$ value relative to the second OH group as commonly reported in the literature [26,27]. Consequently, the use of a stronger base is being required to fully neutralize the *o*-DHT ligand. However, this $(Li_2)(Li,H)$-*o*-DHT intermediate compound having not being reported to date, we decided to go further to complete our knowledge on this materials family.

### 2.2. Synthesis, Characterizations and Thermal Behavior of $(Li_2)(Li,H)$-o-DHT

Taking into account our preliminary observations, the new series of experiments consisted in using 3 equivalents of MeOLi only to produce the titled compound. A white-grey powder was recovered after reaction (16 h, 30 °C in Ar filled glove box), subsequently washed, and characterized by using several characterization techniques. The presence of unaltered *o*-DHT ligand in the compound was readily confirmed by both liquid $^1$H and $^{13}$C NMR measurements (Figure S1) after a derivatization reaction based on a full reprotonation step using $H_2SO_4$ as performed in former studies [18,21]. The as-produced organic salt was also characterized by thermal analyses (TG-DSC-MS) and by Fourier-transform infrared

spectroscopy (FT-IR). Thermal analyses performed under argon (Figure 2) indicated that the compound is methanol-free since the first weight loss ($\Delta m = 23\%$) occurring beyond 300 °C induced a $CO_2$ evolution (i.e., a thermal degradation of the pristine compound and not a desolvation process). The FT-IR spectrum (Figure S2) confirmed well this hypothesis because the presence of MeOH molecules in the crystal structure should notably induce a broad O-H vibration band at 3300–3200 cm$^{-1}$ [20], which was not visible in the present case. However, a weak vibration band located at $\approx$3700 cm$^{-1}$ was detected, which can be assigned to the remaining OH group in the ligand. Finally, the proposed $C_8H_3Li_3O_6$ chemical formula was confirmed by multi-elemental analyses (C, H, Li), as well as by thermal analysis performed under pure $O_2$ (Figures S3 and S4).

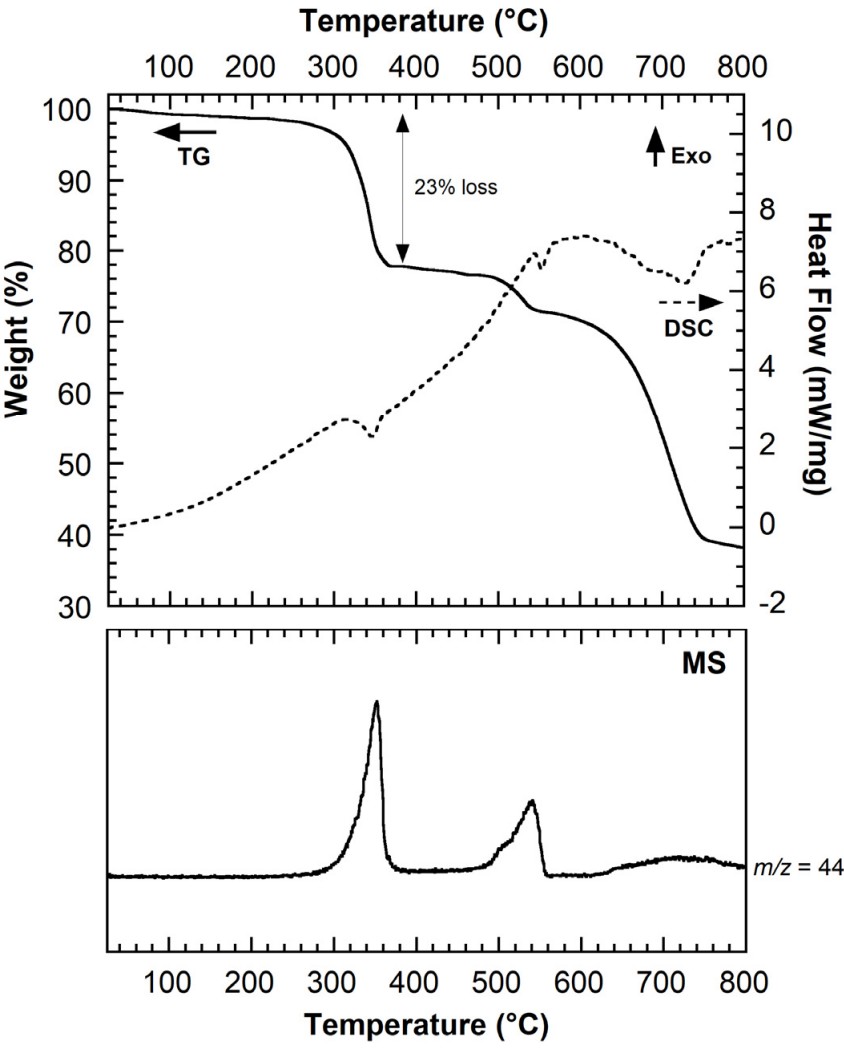

**Figure 2.** Thermal analyses (TG-DSC-MS) of (Li$_2$)(Li,H)-o-DHT recorded under argon at a heating rate of 5 °C.min$^{-1}$ (*m/z* = 44 is ascribed to $CO_2$ evolution).

We decided to further investigate the thermal behavior of (Li$_2$)(Li,H)-o-DHT since the first weight loss observed by TG measurement starting at $\approx$300 °C is associated to an endothermic phenomenon, which is not expected in the case of an irreversible thermal degradation. In fact, we previously experienced that such an endothermic phenomenon accompanied by $CO_2$ release could be ascribed to the formation of a new phase by concerted thermal rearrangement according to the retro-Kolbe–Schmitt reaction [18,21]; this reaction being known for a long time for preparing alkali salts of β-hydroxy acid (BHA) [28]. Therefore, the thermal properties upon heating of (Li$_2$)(Li,H)-o-DHT were studied by temperature-resolved X-ray powder diffraction (TRXRPD) collected from 25 to 500 °C

under $N_2$ at 0.2 $°C.s^{-1}$ (Figure 3). First, the recorded data show that the starting material is crystallized and stable until 265–270 $°C$ in agreement with the thermal analysis data. Beyond this temperature, $(Li_2)(Li,H)$-*o*-DHT undergoes a phase transition with the appearance of new Bragg peaks especially within the 20–30° $2\theta$-range while typical diffraction peaks of the pristine compound vanish. A further heating leads to a complete decomposition process with the progressive formation of crystallized $Li_2CO_3$ with amorphous carbon as expected in such pyrolysis experimental conditions.

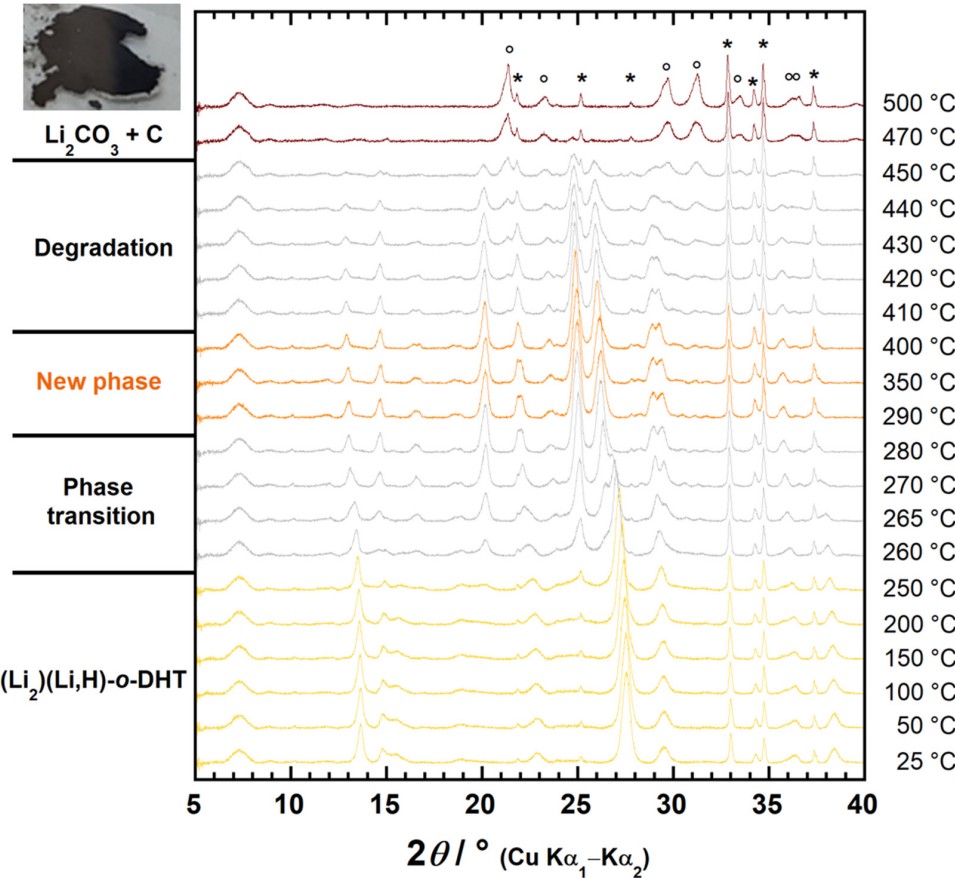

**Figure 3.** TRXRPD patterns collected under nitrogen flow by heating the $(Li_2)(Li,H)$-*o*-DHT compound from 25 to 500 $°C$ (° corresponds to $Li_2CO_3$ diffraction peaks whereas * corresponds to the sample holder). Inset: Photograph of the as-obtained residue after pyrolysis at 500 $°C$ showing a mixture of black and white powders.

With the aim to isolate and characterize this new phase as well as the possible occurrence of a retro-Kolbe–Schmitt thermal reaction, powder of $(Li_2)(Li,H)$-*o*-DHT was heated under controlled pyrolysis conditions and further studied.

### 2.3. Identification of the New Phase and Proposed Thermal Rearrangement Mechanism

To isolate this new phase in sufficient quantity, 600 mg of the $(Li_2)(Li,H)$-*o*-DHT salt was gently annealed at 290 $°C$ during 20 h in a glass oven (Büchi B-585 glassoven Kugelrohr-drying model) placed inside a glove box. As reported earlier [18,21], a thermal gradient can be readily obtained within this glass oven making the separation of the volatile organic compounds (potentially formed) by a simple evaporation/crystallization process possible. Preliminary observations with the naked eyes showed (i) the sample turned from a light gray color into beige upon heating, and (ii) that white and well-shaped small crystals were collected inside the condensation compartment of the glass oven. The latter were readily

identified as pure crystallized catechol by single-crystal XRD analysis (monoclinic $P2_1/c$ crystal structure [28]).

The recovered annealed sample, which had lost 23% in weight, was then characterized by several techniques. While SEM images of the pristine particles of $(Li_2)(Li,H)$-$o$-DHT revealed agglomerations of submicrometer-sized platelets which were a few tenths of a nanometer thick (Figure 4a), the annealed particles exhibited numerous holes and voids (Figure 4b,c) in good agreement with the departure of both $CO_2$ and catechol molecules. Interestingly, liquid $^1H$ and $^{13}C$ NMR measurements after derivatization reaction confirmed again the presence of unaltered $o$-DHT ligand in this new phase (Figure S5). Elemental analyses revealed the presence of 4 Li atoms per $o$-DHT ligand (Figure S6) giving rise to the proposed $C_8H_2Li_4O_6$ chemical formula for this new phase (Figure 1), which formally corresponds to the formation of dilithium (2,5-dilithium-oxy)-terephthalate ($Li_4$-$o$-DHT or $(Li_2)(Li_2)$-$o$-DHT). However, at the macroscopic level, the beige color of this annealed sample differs from the greenish-yellow color previously reported for the $Li_4$-$o$-DHT phase [21].

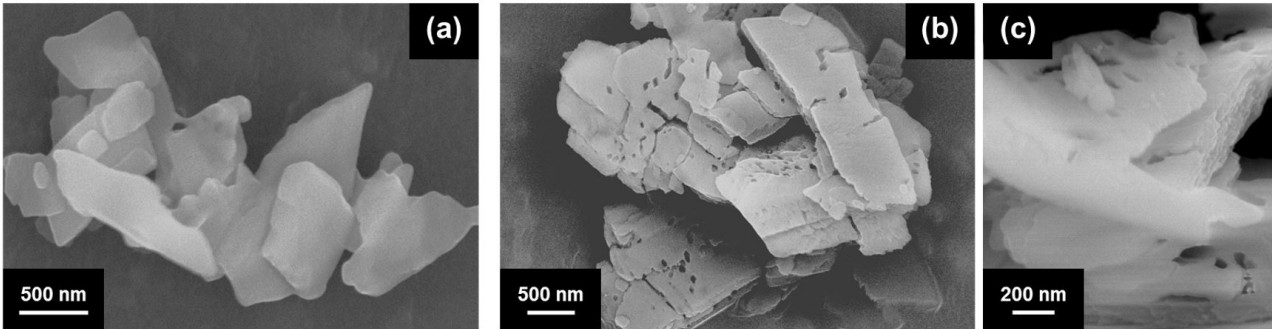

**Figure 4.** SEM images of both the pristine particles of $(Li_2)(Li,H)$-$o$-DHT (**a**) and the recovered powder after thermal treatment at 290 °C during 20 h (**b,c**).

Powder XRD (PXRD) pattern analysis of the recovered sample (Figure 5a) confirmed that the new phase observed by TRXRPD beyond 290 °C (Figure 3) was successfully isolated thanks to this controlled thermal treatment. Beyond the color difference noted above, this collected PXRD pattern did not fit with that previously reported for $Li_4$-$o$-DHT [21] (Figure S7) suggesting that a new polymorph of this tetra-lithiated salt was obtained through this synthesis pathway, hereafter referred to as $Li_4$-$o$-DHT (β-phase). Note that the formerly described form obtained from thermal rearrangement of $(Li_2)(H_2)$-$o$-DHT is now referred as $Li_4$-$o$-DHT (α-phase). The comparison of the FT-IR spectra for these two phases showed many similarities as expected with polymorphs (Figure S8) by displaying the same typical vibrations bands at similar wavenumber values ($\nu_{as}(COO^-)$, $\nu(C=C)$, $\nu_s(COO^-)$ or $\nu(C\text{-}OLi)$), although additional bands were noticed too. In addition, the well-defined PXRD pattern showed that the β-phase exhibits a very good crystallinity compared to the α-phase making further crystallographic analyses possible. After pattern indexing in a monoclinic lattice and space group determination ($P2_1/m$), several attempts were made to solve the crystal structure of the β-phase but we were only able to locate the $o$-DHT unit in the lattice (Figure 5b) and not the four lithium atoms. This is due to the difficulty of finding a coherent oxygen environment around the lithium atoms, which can adopt a wide variety of coordination polyhedra, from highly distorted tetrahedra to regular octahedra [29]. In addition, the presence of a possible preferred orientation in the collected PXRD pattern, suggested by the plate-like crystallites observed by SEM (Figure 4b,c), makes it difficult to fully resolve the structure. Nevertheless, refinement of the cell parameters of the PXRD pattern was successfully achieved in the $P2_1/m$ space group with $a$ = 6.8039(6) Å, $b$ = 13.3934(11) Å, $c$ = 8.8372(8) Å, $\beta$ = 98.415(2)°, and the cell volume $V$ = 796.64(12) Å$^3$ (Figure 5a). The final refinement gives GOF = 1.52 and $R_{wp}$ = 2.41%. Assuming that the formula of the compound is $C_8H_2Li_4O_6$ ($Z$ = 4), the calculated density is 1.85 g·cm$^{-3}$.

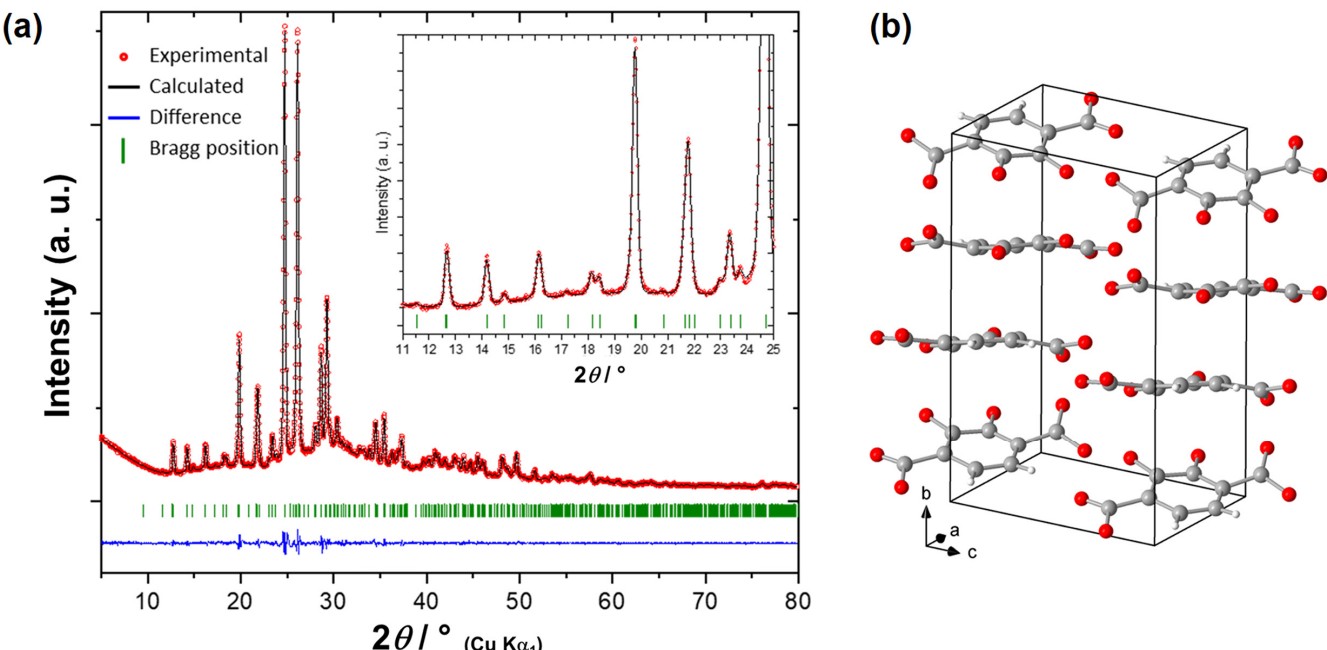

**Figure 5.** (**a**) Le Bail refinement of the PXRD of the Li$_4$-*o*-DHT (β-phase). The inset shows the good agreement between the experimental diffraction peaks and the Bragg positions at the low 2θ angles. (**b**) Location of the *o*-DHT unit (C$_8$H$_2$O$_6$) in the monoclinic lattice as obtained by the FOX software.

Finally, the CO$_2$ evolution observed by TG-DSC-MS (Figure 2) was then quantified in a new series of dedicated experiments using micro gas chromatography. In practice, a few mg of (Li$_2$)(Li,H)-*o*-DHT was placed in a glass ampoule, under argon atmosphere, and heated at 300 °C at 3.5 °C.min$^{-1}$, which is a similar heating rate compared to both thermal analyses and TRXRPD measurements making the control of the pyrolysis reaction possible. The weight loss of the sample due to CO$_2$ release in the gas phase was measured to be 10.9 ± 1.0%. Note that the use of much higher heating rates favors the thermal degradation reaction pathways leading to higher CO$_2$ releases (Figure S9).

By compiling all these experimental results, a peculiar retro-Kolbe–Schmitt reaction seems to occur by controlled pyrolysis of (Li$_2$)(Li,H)-*o*-DHT to produce a new polymorph of Li$_4$-*o*-DHT. Scheme 2 summarizes the proposed thermal concerted rearrangement reaction that involves 4 equivalents of (Li$_2$)(Li,H)-*o*-DHT.

The next study was then focused on the electrochemical storage properties of the β-phase vs. Li to compare its performance towards the α-phase previously reported [21].

### 2.4. Electrochemical Behavior of Li$_4$-o-DHT (β-Phase) as Active Electrode Material vs. Li

The chemical stability of the β-phase in air was first assessed by FT-IR spectroscopy thanks to a series of spectra recorded at different exposure times (Figure S10). While the α-phase oxidizes rapidly in air (solid-state autoxidation reaction [21]), the β-form appears stable with no obvious color change. The main modification of the FT-IR spectra deals with the moisture uptake as shown by the rapid O-H stretching band appearance located in the 3000–3500 cm$^{-1}$ region. This stability in contact with dioxygen implies that the new polymorph could electrochemically desinsert/insert Li ions in the solid state at more than 3 V vs. Li$^+$/Li [18,21]; it should be recalled that an average operating potential limited to 2.85 V vs. Li$^+$/Li was recorded for the α-phase.

**Step 1: Intramolecular decarboxylation (with potential formation of superbasic dicarbanion)**

**Step 2: Intermolecular rearrangement with proton donation from 3 phenol groups**

**Overall reaction:**

$$4\ C_8H_3Li_3O_6 \xrightarrow{\ \Delta\ } 3\ C_8H_2Li_4O_6\ +\ C_6H_6O_2\ +\ 2\ CO_2$$

gas release, total weight loss: ~23 wt%

**Scheme 2.** Proposed thermal concerted retro-Kolbe–Schmitt reaction involving $(Li_2)(Li,H)$-*o*-DHT by controlled pyrolysis reaction to produce $Li_4$-*o*-DHT (β-phase). The by-products, catechol and carbon dioxide, are released in the gas phase. Note that heterolytic cleavage of bonds have been considered for the proposed mechanism without direct evidence.

The electrochemical storage properties of the new polymorph as active electrode material were performed vs. Li in Swagelok®-type cells according to a similar procedure to that reported for the α-phase testing [21] to make comparisons easier. After a preliminary screening, typical electrochemical measurements consisted of galvanostatic cycling tests within the 3.4−2.8 V vs. $Li^+/Li$ potential range recorded at a cycling rate of one $e^-/Li^+$ exchanged per $Li_4$-*o*-DHT unit in 10 h (denoted 1 $Li^+$/10 h).

The most relevant electrochemical features related to $Li_4$-*o*-DHT (β-phase) are summarized in Figure 6 and compared to former data recorded for the α-phase [21]. Figure 6a shows the typical potential-composition ($E - x$) trace restricted to the first five cycles. Note that the use of 1 M $LiClO_4$ in propylene carbonate (PC) as the electrolyte was preferred in this study since slightly better cycling performance was obtained compared to the typical 1 M $LiPF_6$ in ethylene carbonate (EC)/dimethyl carbonate (DMC) (1:1 *vol./vol.*) formulation (Figure S11). Unexpectedly, this new polymorph exhibits a reversible electrochemical deinsertion/insertion of $Li^+$ mostly characterized by a flat voltage plateau (biphasic process) like olivine $LiFePO_4$ or spinel $Li_4Ti_5O_{12}$ and not by a succession of plateaus/solid-solution domains as observed with the α-phase. This two-phase transition occurs at an average working potential of 3.1 V vs. $Li^+/Li$ (compared to 2.85 V vs. $Li^+/Li$ for the α-phase) in agreement with the stability of the β-phase in air underlined above. However, the overall electrochemical process is impeded by a large charge/discharge potential difference ($\Delta E \approx 120$ mV). This polarization value, coupled with the appearance of overpotential activation peaks [30,31] at the beginning of the plateau both in charge and in discharge (Figure 6a, arrows), suggests a sluggish biphasic transition possibly rooted in the existence of high nucleation barrier of the biphasic reaction. This can explain the rapid capacity decrease upon the ten first cycles cycling (35% loss) prior to reaching a stabilized value of $\approx 75$ mAh.g$^{-1}$ over several dozen cycles (Figure 6b). The evolution of the apparent cell resistance ($R_{app.}$) was also monitored upon cycling but no significant change was noticed in agreement with the as-observed stable capacity retention (Figure S12). A cycling test at 60 °C was also performed with the aim to improve the two-phase transition kinetics.

Results indicated just a slight decrease of the polarization value ($\Delta E \approx 100$ mV) but accompanied by a faster capacity decay and a poor coulombic efficiency probably due to the occurrence of side reactions with the electrolyte at such a temperature (Figure S13).

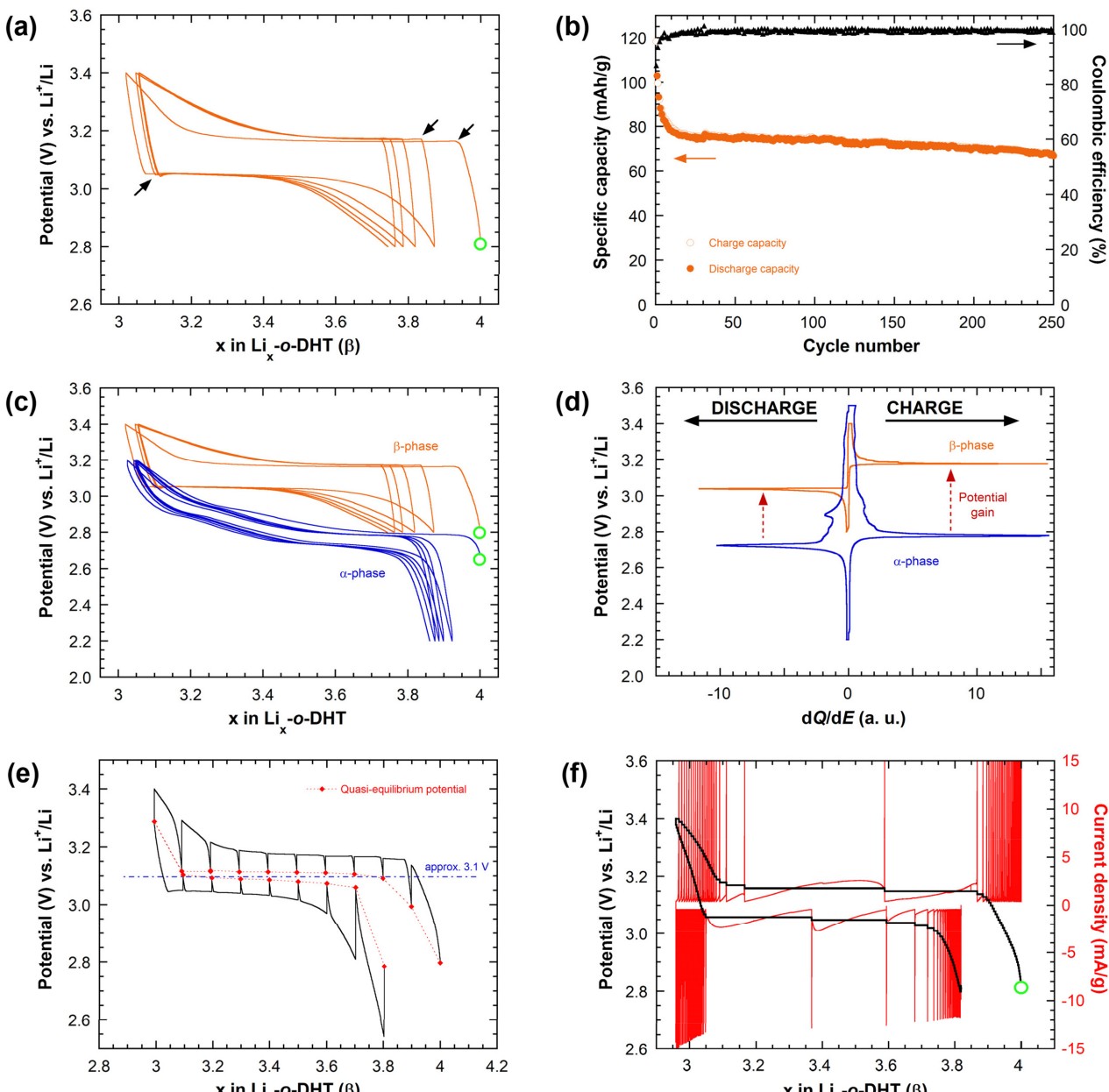

**Figure 6.** (**a**) First five cycles of a Li half-cell using Li$_4$-o-DHT ($\beta$-phase) as the positive electrode material (carbon additive: 33 wt% KB600) galvanostatically cycled at 1 Li$^+$/10 h ($I$ = 10.5 mA.g$^{-1}$). Arrows highlight the overpotential activation peaks at the beginning of each two-phase transition. (**b**) Corresponding capacity retention curve (in orange), together with the coulombic efficiency (in black). (**c**) Superimposition of the potential-composition curves for the two polymorphs using the published data for the $\alpha$-phase (adapted with permission from [21]. Copyright 2014 American Chemical Society). (**d**) Corresponding potential vs. differential capacity curves (first cycle). The red arrow marks the positive potential shift. (**e**) Potential-composition curve for a Li half-cell using Li$_4$-o-DHT ($\beta$-phase) as the positive electrode material (carbon additive: 33 wt% KB600) cycled in GITT mode at 1 Li$^+$/10 h for $\Delta x$ = 0.1 followed by a relaxation period of 48 h. (**f**) PITT curves for a similar cell using potential steps of $\Delta E$ = 10 mV with a current limit corresponding to a rate of 1 Li$^+$/300 h ($I_{min}$ = 0.4 mA.g$^{-1}$). [Electrolyte: 1 M LiClO$_4$/PC; the green circle indicates the starting potential].

The electrochemical discrepancies between the two polymorphs can be more accurately visualized on Figure 6c,d, especially the positive potential shift for the β-phase estimated to +250 mV. Note that the sole reversible and sharp peak observed on the potential vs. differential capacity curve (Figure 6d) is the typical expected trace for a biphasic process. Galvanostatic intermittent titration technique (GITT) was also exploited to get further insight about the as-observed polarization phenomena. The analysis of chronopotentiograms recorded during the open circuit potential (OCP) periods revealed that relatively long relaxation times were necessary to be closer to thermodynamic equilibrium confirming the sluggish biphasic transition. Figure 6e shows the $E - x$ trace measured in GITT mode with an OCP period of 48 h. Basically, the symmetric profile of the GITT curve indicated similar kinetic limitations both in charge and in discharge. Therefore, the equilibrium redox potential value of 3.1 V vs. Li$^+$/Li was verified. The biphasic electrochemical deinsertion/insertion process was further demonstrated by potentiostatic intermittent titration technique (PITT) measurements (Figure 6f). As expected, the reversible flat voltage plateau is better defined by using this electrochemical method. More importantly, the current responses are characterized by bell-shaped curves with roughly symmetric current flow patterns upon charge and discharge meaning that the electrochemical process is not diffusion limited (Cottrellian behaviors) except at the very beginning and at the very end of the overall electrode process, as usual [32]. The related coulometric measurements demonstrate again that the reversible capacity is still limited to half of the theoretical value ($\Delta x = 1$, $Q = 120$ mAh.g$^{-1}$) in our experimental conditions as observed in all our former studies on lithiated DHT-based electrode materials [18,21,23]. Although beyond the scope of this study, the electroactivity of the carboxylate functional groups was also assessed by galvanostatic cycling within the 0.05–2.0 V vs. Li$^+$/Li potential range (Figure S14). A large irreversible capacity was noticed during the first cycle as commonly observed with carboxylate moieties, which can be ascribed to the formation of a solid electrolyte interphase (SEI) [13]. The subsequent plateau that involves the carboxylate lithiation process was found poorly reversible with a rapid capacity loss upon cycling. However, no optimization was performed to improve this electrochemical behavior.

### 2.5. Redox Potential Tuning in the Solid State: From Polymorphism to Through-Space Charge Modulation

At this stage, it should be recalled that the relation between polymorphism and electrochemical insertion properties is well-documented in the literature, especially for inorganic electrode materials [33]. The key component is the crystal arrangement that differs from one form to another inducing variation in terms of ion diffusivity inside the crystal packing and/or electronic structure. For instance, the three TiO$_2$ polymorphs (anatase, rutile, and brookite) distinguish themselves by their Li-insertion properties leading to different polarization and working potentials (1.75, 1.8, and 1.6 V vs. Li$^+$/Li, respectively) [34]. For Nb-based oxides, important gaps of capacity were reported between T-Nb$_2$O$_5$ and B-Nb$_2$O$_5$ (190 mAh.g$^{-1}$ vs. 15 mAh.g$^{-1}$) or between tetragonal and hexagonal LiNbWO$_6$ (200.4 mAh.g$^{-1}$ vs. 20.5 mAh.g$^{-1}$) [35,36]. Tarascon and co-workers [37] have also demonstrated that LiFeSO$_4$F (tavorite-type structure) displays an average operating potential of 3.6 V vs. Li$^+$/Li, whereas the triplite-type polymorph reacts at 3.9 V vs. Li$^+$/Li (i.e., +300 mV as potential gain). However, similar studies on organic electrode materials are scarce to date. Recently, Vlad and co-workers [38] reported the electrochemical behavior vs. Li of a three-dimensional polymorph of the aforementioned Mg(Li$_2$)-*p*-DHT, first reported by our group as a 2-D material [23]. Described in this case as a metal-organic framework (MOF) in the archetypical MOF-74 and referred to as Li$_2$-Mg-DOBDC (DOBDC$^{4-}$ = 2,5-dioxido-1,4-benzenedicarboxylate = *p*-DHT$^{4-}$), such a 3-D structure leads to very different electrochemical features with a large polarization effect and poor cycling performance. With the discovery of Li$_4$-*o*-DHT (β-phase), this new study also underlines the critical importance of the crystal packing for organic electrode materials, as demonstrated with inorganic counterparts for long time.

This situation calls for further efforts in terms of crystal chemistry to get high-voltage lithiated organic electrode materials and, thus, fabricate high-energy full organic or hybrid Li-ion batteries. Basically, the research efforts that have been made for the last twenty years were (logically) focused on organic molecular approaches [7], and notably on the redox potential tuning by playing with substitutions and other (intramolecular) electronic effects [22,39,40], or, in other words, thanks to the traditional "through bond" rationalization without any three-dimensional geometrical consideration concerning these bonds [22]. It was, in fact, our motivation when switching from the $Li_4$-*p*-DHT electrode material to the *ortho*-regiosiomer counterpart, which enabled +300 mV as potential gain with the synthesis of the α-phase [21]. However, the crystal arrangement of the redox-active organic molecular units in the solid state can potentially do more and must be further addressed in the future by the community. In fact, the crystal framework can mitigate or enhance such intramolecular "through bond" electronic effects by "through-space" electrostatic modulation of charge distribution depending on the molecular conformation in the crystal packing [41]. A relevant example has been recently reported by Siew et al. [42] in the field of organic batteries with the tetralithium 2,5-dihydroxy-1,4-benzenediacetate that exhibits a working potential of 3.35 V vs. $Li^+$/Li in the solid phase against 2.70 V measured in solution. Although the respective crystal structures are not known, it can be hypothesized that the potential gain observed when switching from $Li_4$-*o*-DHT (α) to $Li_4$-*o*-DHT (β) is related to conformational considerations. As a result, the cumulative potential gain that exceeds half a volt observed between $Li_4$-*p*-DHT and $Li_4$-*o*-DHT (β) combine the two effects: "through bond" and "through-space" electrostatic modulation of charge distributions.

## 3. Materials and Methods

**General analytical techniques and characterizations:** Liquid $^1$H and $^{13}$C NMR spectra were recorded on Bruker AVANCE III at 400 MHz and 100 MHz, respectively. Chemical shifts (δ) are given in ppm relative to TMS. DMSO-$d_6$ was purchased from Eurisotop Company (purity > 99.5%). For the derivatization reaction, a few drops of $H_2SO_4$ were added to the *o*-DHT-based compounds in a sealed NMR sample tube in a glove box. Elemental analyses (C, H) were obtained by combustion analyzer using either a Flash 2000 CHN analyzer (Thermo Scientific) for $H_4$-*o*-DHT or an UNICUBE analyzer (elementar) for the corresponding lithiated salts. The Li content was precisely determined by Inductively Coupled Plasma-Atomic Emission Spectroscopy (ICP-AES) thanks to an iCAP 6300 radial analyzer (Thermo Scientific). A mono-element solution of Li (1000 ppm, CHEMLAB) was used for the calibration. Fourier-transform infrared spectroscopy measurements were recorded on a FT-IR Vertex 70 (Bruker), and pellets were prepared by mixing the synthesized materials with spectroscopic-grade potassium bromide at 0.5% (*w/w*). The thermogravimetric-differential scanning calorimetry (TG-DSC) analyses were performed with a SENSYS evo (Setaram) whereas thermal analyses coupled with mass spectrometry were performed with a STA449F3 Jupiter and QMS403C Aëolos instruments (NETZSCH). Scanning electron microscopy (SEM) images were collected on a JSM-7600F microscope (JEOL).

**X-rays diffraction techniques and refinements:** Temperature-resolved X-Ray Powder Diffraction (TRXRPD) experiments were performed on a D8 Advance diffractometer (Bruker) equipped with an Anton Parr Chamber HTK 1200 N high-temperature attachment. Data were collected in Bragg-Bentano geometry with a Cu-anode X-ray source operated at 40 kV and 40 mA. The Cu-K$\beta$ radiation was filtered by means of a Ni foil. The experiment was performed under nitrogen flow with a step of 0.016724° and an acquisition time of 1.2 s per step (heating rate of 0.2 °C·s$^{-1}$). Single crystal X-ray diffraction measurement were performed at room temperature using a XtaLAB Synergy diffractometer (Rigaku) operating with Mo-K$\alpha$ radiation (λ = 0.71073 Å). The powder XRD (PXRD) patterns were typically recorded in air at room temperature with a Bruker D8 diffractometer operating with Cu-K$\alpha_1$ radiation (λ = 1.54060 Å) and a LynxEye detector in Bragg-Brentano geometry except for capillary powder samples. In the latter case, PXRD patterns were collected in transmission mode on an INEL XRG CPS120 diffractometer operating with Cu-K$\alpha_1$ radia-

tion ($\lambda$ = 1.54060 Å) and equipped with a curved detector, ranging from 0 to 120°. Pattern indexing, space group determination and structure solution by direct space methods using parallel tempering algorithm and Monte Carlo global optimization were performed in FOX software [43]. Le Bail refinements were performed in Jana2006 [44] using a pseudo-Voigt function for the peak profile. The asymmetry of the peaks was corrected with the Berar–Baldinozzi function [45].

**CO$_2$ measurement:** In order to quantify the carbon dioxide release, 5 mg of (Li$_2$)(Li,H)-$o$-DHT was placed in a glass ampoule, under argon atmosphere, and heated at 300 °C at 3.5 °C.min$^{-1}$. The gaseous atmosphere was then analyzed with a micro-gas chromatography set-up ($\mu$-GC-R3000, SRA instrument) using ultra-high purity (99.9999%) argon and helium carrier gases. Prior to the acquisition of the chromatogram, the gas contained in the ampoule was mixed with argon at 1.5 bar for five minutes. Various gases can be identified and quantified (H$_2$, O$_2$, N$_2$, CH$_4$, CO, and CO$_2$). Only CO$_2$ was identified in the present work. It was then quantified. The response of the micro-gas chromatography towards CO$_2$ is linear in a wide concentration range. The estimated error in the gas measurement was less than 10%.

**Synthesis Procedures**: 1,2-dihydroxybenzene (catechol or pyrocathecol) was supplied by Solvay, potassium hydrogen carbonate (99%) by Sigma-Aldrich, dry ice by Cryo'Ice, concentrated hydrochloric acid (37 wt%) was purchased from Fisher Scientific. Lithium methoxide (10% in MeOH) and anhydrous MeOH were purchased from Sigma-Aldrich.

*2,3-Dihydroxyterephthalic acid (H$_4$-o-DHT):* 1,2-Dihydroxybenzene (2.00 g, 18.2 mmol) and potassium hydrogen carbonate (5.89 g, 58.8 mmol, 3.23 eq.) were ground with a mortar and pestle then placed in a Parr reaction system (40 mL) with dry ice (11.69 g, 265.7 mmol, 14.6 eq.) to reach 20 bars. The Parr reactor was then heated overnight to 225 °C leading to a final auto-generated pressure of 45 bars. The recovered crude material was suspended in diethyl ether (50 mL) for 30 min then filtrated. The dark green powder was dissolved in 1L of water for one hour and filtrated on Celite 535. Eventually, concentrated HCl was added to the liquid to precipitate 2,3-dihydroxyterephthalic acid. After filtration on Millipore PVdF, the creamy beige precipitate was dried under vacuum at 120 °C for 24 h resulting in pure H$_4$-o-DHT (2.60g, 72%). $^1$H NMR (400 MHz, DMSO-d$_6$) $\delta$ (ppm) 7.28 (s, 2H, H arom.); $^{13}$C NMR (100 MHz, DMSO-d$_6$) $\delta$ (ppm): 171.5 (COOH), 150.7 ($C-$OH), 118.5 ($C-$COOH), 116.7 (CH); IR (KBr pellet): $\nu_{max}$/cm$^{-1}$ 3523-2525 ($\nu$ O$-$H, $\nu$ C$-$H), 1949, 1659 ($\nu$ C=O), 1622, 1569, 1486-1396 ($\nu$ C=C), 1346, 1323, 1227-1170 ($\nu$ C$-$O, $\nu$ OC$-$OH, $\nu$ O$-$H), 889-753 ($\nu$ C$-$H), 660, 559, 493, 432. Elemental Analysis: %C = 48.54% (th. 48.50%); %H = 3.17% (th. 3.05%).

*Dilithium (3-hydroxy-2-lithium-oxy)terephthalate ((Li$_2$)(Li,H)-o-DHT)*: In a Ar-filled glove box with H$_2$O < 0.1 ppm and O$_2$ < 0.1 ppm, a solution of lithium methoxide (10% in methanol, 2.2 M, 3.35 mL, 75.0 mmol, 3 eq.) was added dropwise to a solution of H$_4$-o-DHT (495 mg, 2.5 mmol) in methanol (25 mL). The dark-brown solution was stirred overnight to allow Li$_3$H-o-DHT precipitation. The white-grey solid was collected by centrifugation, washed repeatedly by shaking with anhydrous diethyl ether, and dried in vacuum at 50 °C for 1 h (389.9 mg, yield 72%). Due to its insolubility in common NMR solvents, the formation of Li$_3$H-o-DHT was confirmed after a derivatization reaction based on a full reprotonation step: 4 drops of concentrated H$_2$SO$_4$ were added to a heterogeneous solution of Li$_4$H-o-DHT (7 mg) in DMSO-d$_6$ (1 mL) resulting in a homogeneous brown-colored solution. Both $^1$H and $^{13}$C NMR spectra correspond to pure 2,3-dihydroxyterephthalic acid. $^1$H NMR (400 MHz, DMSO-d$_6$ with drops of H$_2$SO$_4$) $\delta$ (ppm) 12.12 (s, H$_3$O$^+$), 7.18 (s, 2H, C$_{Ar}$-H); $^{13}$C NMR (100 MHz, DMSO-d6 with drops of H$_2$SO$_4$) $\delta$ (ppm) 171.3 (COOH), 150.5 (C-OH), 118.5 (**C**-COOH), 116.6 (CH); IR (KBr pellet): $\nu_{max}$/cm$^{-1}$ 3010 ($\nu$ O-H, $\nu$ C-H), 1648 ($\nu_{as}$ OC-OLi), 1479-1462 ($\nu$ C=C), 1398 ($\nu_s$ OC-OLi), 1274 ($\nu$ C-OLi), 1220 ($\nu$ O-H), 1146, 1020, 945, 896, 842-662 ($\nu$ C-H), 571, 517, 470, 412. ICP-AES: Theoretical Li = 9.64%; Experimental: Li = 9.35% ± 0.57. Elemental Analysis: Theoretical C = 44.50%; H = 1.40%. Experimental: C = 43.43%; H = 1.74%.

*Dilithium (2,3-dilithium-oxy)-terephthalate ((Li$_2$)(Li$_2$)-o-DHT orLi$_4$-o-DHT, β phase):* In a glove box, dilithium (3-hydroxy-2-lithium-oxy)terephthalate (typically 200 mg, 0.92 mmol) was ground with a mortar and pestle then placed in a glass oven (Büchi B-585 glassoven Kugelrohr-drying model). Note that the glass oven compartment was initially evacuated (15 mbar) then isolated before heating. The powder was then heated up to a real temperature of 293 °C (300 °C on the display) for 20 h. This thermal treatment produces a beige powder of pure β-Li$_4$-o-DHT (154 mg, quantitative yield) with both catechol and $CO_2$ releases. $^1$H NMR (400 MHz, DMSO-d$_6$ with drops of $H_2SO_4$) δ (ppm): 12.08 (s, $H_3O^+$), 7.21 (s, 2H, C$_{Ar}$-H); $^{13}$C NMR (100 MHz, DMSO-d$_6$ + $H_2SO_4$) δ (ppm): 171.3 (COOH), 150.5 (C-OH), 118.6 (**C**-COOH), 116.62 (CH); IR (KBr pellet): $\nu_{max}/cm^{-1}$ 1543 ($\nu_{as}$ OC-OLi), 1456 ($\nu$ C=C), 1396 ($\nu_s$ OC-OLi), 1294, 1241 ($\nu$ C-OLi), 1147, 1020, 966, 862-777 ($\nu$ C-H), 671, 617, 575, 565, 509, 456. ICP-AES: Theoretical Li = 12.51%; Experimental: Li = 12.22 ± 0.58%. Elemental Analysis: Theoretical C = 43.31%; H = 0.91%. Experimental: C = 42.69%; H = 1.37%.

**Electrochemical study:** The electrochemical performance of the materials was tested vs. lithium in Swagelok®-type cells using a Li metal disc as the negative electrode and fiberglass separators (Whatman®) soaked with 1 M LiClO$_4$ in propylene carbonate (PC) as the electrolyte unless otherwise noted. The PC solvent and the common "LP30" battery grade electrolyte (i.e., LiPF$_6$ 1 M in EC:DMC 1:1 *vol./vol.*) were directly employed as received from Novolyte. The positive electrodes were prepared without binder in an argon-filled glove box by grinding a powder of Li$_4$-o-DHT (β-phase) with 33 wt% carbon black (Ketjenblack EC-600JD, AkzoNobel denoted KB600 or C-NERGY SUPER C65, IMERYS denoted C65) with a pestle in a mortar. The typical positive electrode weight was ≈3.5 mg. The electrochemical measurements were performed by using a MPG system (Bio-Logic S.A., Seyssinet-Pariset, France). The apparent cell resistance (R$_{app.}$), also known as internal cell resistance, was determined by the simple current interrupt technique by using the open circuit voltage (OCV) periods integrated in the cycling program at each half-cycle.

## 4. Conclusions

In this complementary study focused on the *ortho*-regioisomer backbone of the DHT ligand, a novel partially lithiated intermediate (Li$_2$)(Li,H)-o-DHT compound was isolated and characterized. Its controlled thermal pyrolysis allowed preparing a polymorph (β-phase) of the formerly described Li$_4$-o-DHT compound (α-phase). The electrochemical investigations of this new polymorph revealed an electrochemical activity vs. Li still limited to half of the theoretical capacity in our experimental conditions. Unfortunately, the electrode reaction was based on a sluggish biphasic transition process inducing a polarization effect ($\Delta E \approx 120$ mV) combined with limited capacity recovery. Although the β-phase can still sustain a reversible capacity value of ≈75 mAh.g$^{-1}$ over several dozen cycles following the first ten cycles, the overall electrochemical performance was judged lower than that observed with the α-phase. Nevertheless, this study was the occasion to notice a positive potential shift for this polymorph estimated to +250 mV compared to the α-phase, which demonstrates the critical importance of the crystal arrangement by mitigating intramolecular electronic effects by through-space charge modulation. This finding paves the way for further research by smartly tuning electronic effects at molecular level as well as in the solid state or in other words, by combining molecular engineering (organic chemistry) and crystal design (materials science).

**Supplementary Materials:** The following supporting information can be downloaded at: https://www.mdpi.com/article/10.3390/inorganics10050062/s1, Figure S1: Typical 1H NMR (a) and 13C NMR (b) spectra of (Li$_2$)(Li,H)-o-DHT measured in $H_2SO_4$/DMSO-d6 solution (derivatization reaction); Figure S2: Typical FT-IR spectrum of (Li$_2$)(Li,H)-o-DHT (KBr pellet); Figure S3: Rectilinear calibration graphs for the lithium quantification in (Li$_2$)(Li,H)-o-DHT measured by ICP-AES considering two atomic emission lines for Li (610.3 nm and 670.7 nm, respectively); Figure S4: (a) Thermal analyses (TG-DSC) of (Li$_2$)(Li,H)-o-DHT performed under pure O$_2$ at a heating rate of 5 °C.min$^{-1}$. (b) PXRD pattern of the residue recovered after thermal annealing of (Li$_2$)(Li,H)-o-DHT at 1000 °C

under pure $O_2$ (combustion) and measured in a quartz capillary due to the small amount of recovered powder; Figure S5: Typical 1H NMR (a) and 13C NMR (b) spectra measured in $H_2SO_4$/DMSO-d6 solution (derivatization reaction) of the recovered powder (new phase) after thermal treatment of $(Li_2)(Li,H)$-*o*-DHT at 290 °C during 20 h; Figure S6. Rectilinear calibration graphs for the lithium quantification of the recovered powder after thermal treatment of $(Li_2)(Li,H)$-o-DHT at 290 °C during 20 h by ICP-AES considering two atomic emission lines for Li (610.3 nm and 670.7 nm, respectively); Figure S7. Overlaid of the representative PXRD patterns for $Li_4$-*o*-DHT ($\alpha$-phase) in green and Li4-o-DHT ($\beta$-phase) in orange extracted from TRXRPD measurements at 245 °C [14] and 290 °C (Figure 3 in the main text), respectively (* corresponds to the sample holder); Figure S8. Typical FT-IR spectrum (KBr pellet) of the recovered powder (new phase) after thermal treatment of $(Li_2)(Li,H)$-o-DHT at 290 °C during 20 h (in orange); the as-obtained phase is ascribed to $Li_4$-*o*-DHT ($\beta$-phase); Figure S9. Chromatogram measured after heating of $(Li_2)(Li,H)$-o-DHT to 300 °C at a heating rate of 60 °C.min$^{-1}$; Figure S10. Evolution of FT-IR spectra of Li4-o-DHT ($\beta$-phase) after air exposure; Figure S11. First five cycles of a Li half-cell using $Li_4$-*o*-DHT ($\beta$-phase) as the positive electrode material (carbon additive: 33 wt% KB600) galvanostatically cycled at 1 Li$^+$/10 h as reported in Figure 6a in the main text by using this time the common "LP30" battery grade electrolyte (i.e., LiPF6 1 M in EC:DMC 1:1 vol./vol.); Figure S12. Capacity retention curve upon cycling (from Figure 6b in the main text) together with the corresponding evolution of the apparent cell resistance measured in charge and reported every ten cycles. [Electrolyte: 1 M LiClO$_4$/PC]; Figure S13. First six cycles of a Li half-cell using $Li_4$-*o*-DHT ($\beta$-phase) as the positive electrode material (carbon additive: 33 wt% KB600) galvanostatically cycled at 1 Li$^+$/10 h at 60 °C. [Electrolyte: 1 M LiClO$_4$/PC; the green circle indicates the starting potential]; Figure S14. First five cycles of a Li half-cell using $Li_4$-*o*-DHT ($\beta$-phase) as the positive electrode material (carbon additive: 33 wt% C65) galvanostatically cycled at 1 Li$^+$/10 h (first discharge at 1 Li$^+$/5 h). [Electrolyte: 1 M LiClO4/PC; the green circle indicates the starting potential].

**Author Contributions:** Conceptualization, P.P. and S.R.; methodology, L.B., A.J., E.Q., Y.L.-S., S.L.C., S.R. and P.P.; validation, E.Q., S.L.C., S.R. and P.P.; formal analysis, L.B., A.J., E.Q., Y.L.-S., S.L.C., S.R. and P.P.; investigation, L.B., A.J., E.Q., Y.L.-S. and P.P.; writing—original draft preparation, L.B., E.Q., S.L.C., S.R. and P.P.; writing—review & editing, L.B., A.J., E.Q., Y.L.-S., S.L.C., S.R. and P.P.; supervision, S.R., S.L.C., P.T.-V. and P.P.; funding acquisition, P.T.-V. All authors have read and agreed to the published version of the manuscript.

**Funding:** This work was funded by the National Agency for Research and Technology (ANRT) through an Industrial Research Convention (CIFRE) between IMN/UMR 6502 and Renault group (n°2018/1264).

**Institutional Review Board Statement:** Not applicable.

**Informed Consent Statement:** Not applicable.

**Data Availability Statement:** All the data are available within the manuscript or the supporting information.

**Acknowledgments:** PP would like to dedicate this article to Michel Armand, one of the world's pioneering and leading scientists in the field of secondary batteries, for the long-standing collaborative work in the field of organic batteries. This work includes NMR experiments carried out on the CEISAM NMR platform, Nantes Université. Elemental analyses (C, H) were performed both at the CEISAM laboratory (D. Loquet), Nantes Université, and at the CEA-Liten (M. Nicolas), Grenoble. The Li analyses by ICP-AES were performed at the LPG-UMR 6112 (C. La, M. Rivoal), Nantes Université.

**Conflicts of Interest:** The authors declare no conflict of interest.

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
