# Peer review of "Influence of Polymorphism on the Electrochemical Behavior of Dilithium (2,3-Dilithium-oxy)-terephthalate vs. Li"

_inorganics, doi:10.3390/inorganics10050062_

Round 1
Reviewer 1 Report
In this work, the authors reported a new organic cathode based on two lithium oxide groups at ortho position. Unlike the other organic cathodes, Li4-o-DHT has a lithium-rich structure, which could be used to couple with the normal lithium battery anodes such as graphite, etc. In addition, the material was well characterized by several techniques including XRD, FTIR, NMR, TGA, SEM, and elemental analysis. In lithium batteries, it shows a pair of high redox plateaus above 3.0V and a stable cycle life of 250 cycles, demonstrating a promising organic cathode. The synthetic part is very important and inspiring for further development of Li-rich organic cathodes. I think this work is very important for the development of organic cathodes, and I recommend it for publication after minor revision.
- Since there are two carboxylate groups at the para position of the benzene ring, this organic material should also be able to act as an organic anode. The electrochemical behavior of Li4-o-DHT in a low cutoff window from 0.1 to 2V should be provided.
- The impedance tests for Li4-o-DHT before and after cycling is recommended to monitor the impedance evolution upon cycling.
- Some explanation should be provided to clarify why 1 M LiClO4 in propylene carbonate (PC) is better than 1 M LiPF6 in ethylene carbonate/dimethyl carbonate as the electrolyte for Li4-o-DHT?
Reviewer 2 Report
The authors presented an interesting paper with an accurate and detailed description of the synthesis process and study of a new Li4-o-DHT polymorph. The article may be published after minor corrections. The following issues could be taken into consideration before preparing the final version of the manuscript:
1) Most likely, the high overvoltage during the charge/discharge of the β-phase did not allow obtaining the full (de)intercalation capacity of the material. Most probably, it's possible to diminish the effect of high resistance by slightly heating the cell, for example, up to +50 or +75oC. A PC-based electrolyte is quite suitable for this type of experiment which can be carried out to improve this article or in subsequent works
2) For the audience of the Inorganics journal, the explanation of the higher potential of the β-phase through “intramolecular electronic effects by through-space charge modulation” may seem not entirely clear and obvious. It is proposed to expand the discussion of the results in this area
